# Central venous pressure and dynamic indices to assess fluid appropriateness in critically ill patients: A pilot study

**Chiara Prezioso, Roberta Trotta, Erika Cavallo, Federica Fusina◉\*, Elena Malpetti, Filippo Albani, Rosalba Caserta, Antonio Rosano, Giuseppe Natalini**

Department of Anesthesia and Intensive Care, Fondazione Poliambulanza, Brescia, Italy

\* f.fusina@gmail.com

## Abstract

### Background

The correct identification of the appropriateness of fluid administration is important for the treatment of critically ill patients. Static and dynamic indices used to identify fluid responsiveness have been developed throughout the years, nonetheless fluid responsiveness does not indicate that fluid administration is appropriate, and indexes to evaluate appropriateness of fluid administration are lacking. The aim of this study was to evaluate if central venous pressure (CVP) anddynamic indices could correctly identify fluid appropriateness for critically ill patients.

### Methods

Data from 31 ICU patients, for a total of 53 observations, was included in the analysis. Patients were divided into two cohorts based on the appropriateness of fluid administration. Fluid appropriateness was defined in presence of a low cardiac index ($< 2.5$ l/min/m$^2$) without any sign of fluid overload, as assessed by global end-diastolic volume index, extravascular lung water index or pulmonary artery occlusion pressure.

### Results

For 10 patients, fluid administration was deemed appropriate, while for 21 patients it was deemed inappropriate. Central venous pressure was not different between the two cohorts (mean CVP 11 (4) mmHg in the fluid inappropriate group, 12 (4) mmHg in the fluid appropriate group, p 0.58). The same is true for pulse pressure variation (median PPV 5 [2, 9] % in the fluid inappropriate group, 4 [3, 13] % in the fluid appropriate group, p 0.57), for inferior vena cava distensibility (mean inferior vena cava distensibility 24 (14) % in the fluid inappropriate group, 22 (16) % in the fluid appropriate group, p 0.75) and for changes in end tidal carbon dioxide during a passive leg raising test (median d.E$_T$CO$_2$ 1.5 [0.0, 2.0]% in the fluid inappropriate group, 1.0 [0.0, 2.0] % in the fluid appropriate group, p 0.98). There was no association between static and dynamic indices and fluid appropriateness.

**Data Availability Statement:** The anonymized data set is available at this address: https://github.com/filippo1985/afadysta.git.

**Funding:** The authors received no specific funding for this work.

**Competing interests:** The authors have declared that no competing interests exist.

**Abbreviations:** AUC of the ROC curve, Area Under the retrieving operative characteristic Curve; BSA, body surface area; CI, cardiac index; CVP, central venous pressure; $d.E_TCO_2$, changes in $E_TCO_2$ during a passive leg raising test; ELWI, extravascular lung water index; $E_TCO_2$, end tidal $CO_2$; $E_TCO_2$, end tidal $CO_2$; EVLW, extravascular lung water; GEDV, global end-diastolic volume; GEDVI, global end diastolic volume index; ICU, Intensive Care Unit; IVC, inferior vena cava; PAC, pulmonary artery catheter; PAOP, pulmonary artery occlusion pressure; PAP, pulmonary artery pressure; PBW, predicted body weight; PiCCO, Pulse index Contour Continuous Cardiac Output; PLR, passive leg raising; PPV, pulse pressure variation; PVPI, pulmonary vascular permeability index.

## Conclusions

Central venous pressure, pulse pressure variation, changes in end tidal carbon dioxide during a passive leg raising test, inferior vena cava distensibility were not associated with fluid appropriateness in our cohorts.

## Background

Fluid administration in critically ill patients has been the object of considerable debate for more than 50 years [1, 2], and fluid administration is suggested by the most recent guidelines as the initial treatment for patients with sepsis and septic shock [3].

Static and dynamic indices have been used in order to evaluate the ability to increase stroke volume in response to fluid administration (i.e. "fluid responsiveness") [4], with dynamic indices performing better than static ones [5–7].

Nonetheless, fluid responsiveness does not indicate that fluid administration is appropriate [8], but only that fluid administration increases cardiac output. In the presence of a normal cardiac output, there is no benefit in increasing it to supranormal values [2, 9]: in fact, it appears to be harmful since excessive fluids might worsen outcome [10–12]. Instead, in patients with low cardiac output and without any signs of pulmonary circulation overload, fluid administration appears to be indicated [8]. Consequently, before asking ourselves if a patient is a fluid responder, we should be asking if fluid administration is appropriate for that patient.

Central venous pressure (CVP), a static index, has turned out to be an inadequate predictor of fluid responsiveness. Nonetheless, to our knowledge, its usefulness in identifying patients in whom fluid administration is appropriate or inappropriate has not been evaluated before. Therefore the aim of this study was to evaluate if CVP and, as a secondary outcome, dynamic indices [pulse pressure variation (PPV), inferior vena cava (IVC) distensibility and changes in end tidal carbon dioxide during a passive leg raising test $(d.E_TCO_2)$], could correctly identify fluid appropriateness.

## Methods

This prospective observational study was performed in the Intensive Care Unit (ICU) of Fondazione Poliambulanza, Brescia, Italy. The protocol was approved by the local ethical committee (Comitato Etico della Provincia di Brescia, protocol number NP 3056). All methods were performed in accordance with the relevant guidelines and regulations and in accordance with the Declaration of Helsinki. Written informed consent was obtained from participants and/or their legal guardians. If it was not possible to obtain informed consent from the patients at time of enrollment due to their clinical condition, it was obtained at a later date, provided that the patients regained the ability to give consent. All data was anonymized.

All patients who were admitted to ICU from December 2018 to December 2019 were screened. Patients were enrolled if they met all of the following criteria: age $\geq$ 18 years; tracheal intubation and controlled mechanical ventilation; central venous catheter and arterial line in place; absence of any sign of spontaneous respiratory activity (evaluated by airway pressure and airflow waveforms); mean arterial pressure < 65mmHg; sinus rhythm; invasive cardiac output measurement using pulmonary artery catheter (PAC, Edwards Lifesciences LLC, USA) or transpulmonary thermodilution with PiCCO device (Pulse index Contour Continuous

Cardiac Output, Pulsion Medical Systems, Germany); hemodynamic stability in the fifteen minutes preceding invasive cardiac output measurement (stable mean arterial pressure without changes in inotropic and vasopressor dose).

Exclusion criteria were intracranial hypertension or cervical and long bone unstable fractures.

## Protocol

Cardiac index (CI), CVP, pulmonary artery occlusion pressure (PAOP), global end diastolic volume index (GEDVI), extravascular lung water index (ELWI), PPV, IVC distensibility and $d.E_TCO_2$ during a passive leg raising test were measured in each patient [13].

Heart rate and mean arterial pressure were continuously recorded. All cardiovascular pressures were referenced at the mid-axillary line [14, 15].

Static and dynamic fluid responsiveness indices were calculated as follows: in patients with PAC, cardiac output was measured by thermodilution and the PAOP and the pulmonary artery pressure (PAP) were recorded.

In patients with transpulmonary thermodilution devices, cardiac output, estimated global end-diastolic volume (GEDV), and extravascular lung water (EVLW) were measured. Pulmonary vascular permeability index (PVPI) was calculated as the ratio between EVLW and pulmonary blood volume, which is calculated as GEDV * 0.25.

CO, GEDV and EVLW were indexed to body surface area (BSA) calculated as BSA in $m^2$ = (weight in $kg^{0.425}$ * height in $cm^{0.725}$) * 0.007184 [16] and to predicted body weight (PBW) calculated with Devine's formula [17] as PBW in kg = 50 + 0.91 (height in cm– 152.4) for males and PBW in kg = 45.5 + 0.91 (height in cm)– 152.4) for females, respectively.

PPV was calculated as the difference between maximum and minimum pulse pressure over a respiratory cycle, divided by their average.

Measurement of $d.E_TCO_2$ during passive leg raising (PLR) test was performed by placing patients in a semi-recumbent position at 45˚, then legs were elevated to 45˚ for 60 seconds [5], end tidal $CO_2$ ($E_TCO_2$) was recorded immediately before and at the end of the PLR test [13].

The ultrasound measurement of distensibility of the IVC was performed by measuring the change in diameter of the IVC (in long axis view M-mode, 2–3 cm distal to the confluence with the hepatic vein) during the respiratory cycle [18]. The IVC variation induced by the respiratory cycle was calculated as the difference between the maximum and the minimum IVC diameter, normalized by the mean of the two values and expressed as a percentage [19].

Measurements could be repeated after an hemodynamic intervention.

## Definition of fluid appropriateness

The appropriateness of fluid administration was evaluated as follows: for patients with CI $\geq$ 2,5 l/min/$m^2$, volemic expansion was **considered inappropriate** [20]. For patients with CI <2.5 l/min/$m^2$, fluid administration was **considered appropriate** [20] in the absence of fluid overload. The absence of fluid overload was identified when PAOP was <18 mmHg [20] in patients with PAC, or, in patients with PiCCO, when GEDVI was <680 ml/$m^2$ or when GEDVI < 800 ml/$m^2$ and ELWI < 10 ml/Kg [21, 22].

If CI <2.5 l/min/$m^2$ and fluid overload was present, fluid administration was **considered inappropriate** [20].

Patients were therefore divided into two cohorts (Fig 1): patients for whom fluid administration was considered appropriate ("**fluid appropriate**" group) and patients for whom fluid administration was considered inappropriate ("**fluid inappropriate**" group).

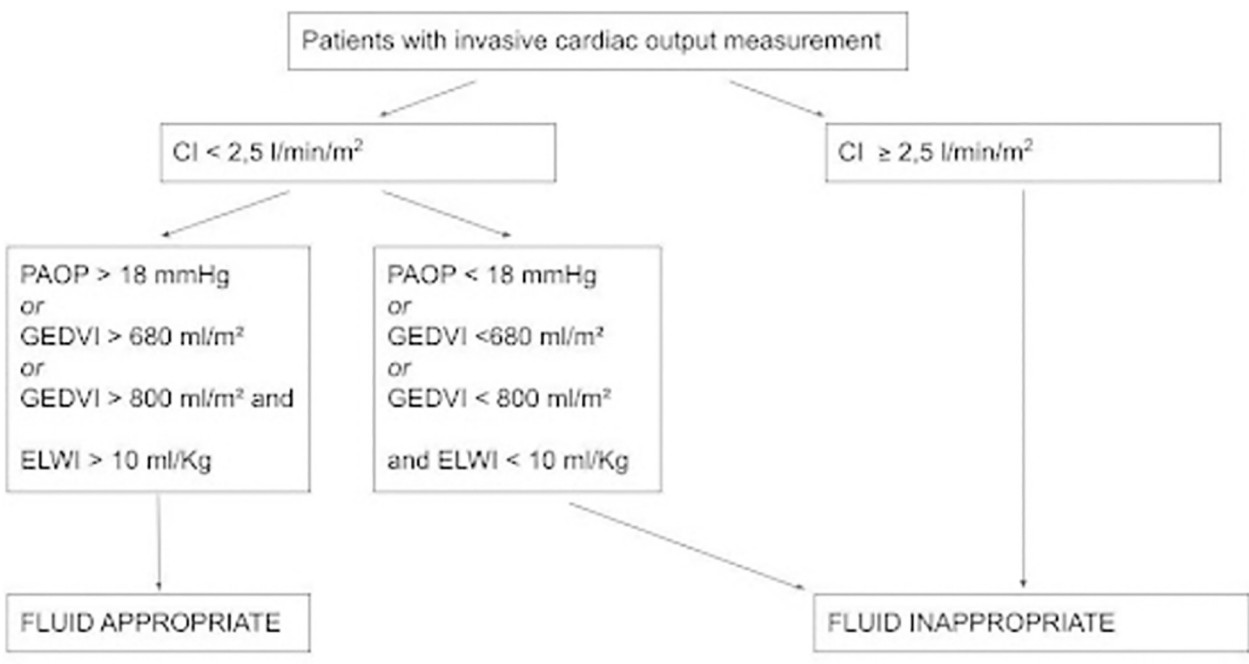

**Fig 1. Definition of fluid appropriateness.** Abbreviations: CI = cardiac index, PAOP = pulmonary artery occlusion pressure, GEDVI = global end diastolic volume index, ELWI = extravascular lung water index.

## Study outcome

The primary outcome of the study was to compare CVP values between the cohort of patients for whom fluid administration was considered appropriate and the cohort of patients for whom fluid administration was considered inappropriate.

Secondary outcome was the comparison of PPV, IVC distensibility and $d.E_TCO_2$ after passive leg raising between the two cohorts.

## Statistical analysis

Data is expressed as count (percentage), mean (standard deviation) or median [1st-3rd quartile], and the comparison between CVP, PPV, IVC distensibility and $d.E_TCO_2$ after passive leg raising between the two cohorts was performed with Student's t-test and Wilcoxon test, as appropriate, therefore only the first measurement recorded for each patient was used, in presence of repeated measurements. The presence of ten patients in each cohort, hypothesizing a mean CVP of 9 mmHg with a standard deviation of 2.2 mmHg [7], would guarantee a power of 0.80 to identify a difference in CVP between the two groups of 3 mmHg, with a type I error frequency of 0.05.

We also analyzed the association between fluid appropriateness and static and dynamic indices levels using linear mixed effect models for each index (CVP, PPV, $d.E_TCO_2$, IVC distensibility). The dependent variable was fluid appropriateness and the age and cardiac index were chosen *a priori* as covariates, with patients as random effect. Repeated measurements were included to evaluate associations with the linear mixed effects model.

A p value lower than 0.05 was considered significant. Statistical analyses were performed with R 3.6.3 (R Core Team, 2021. R Foundation for Statistical Computing, Vienna) [23].

**Table 1.  Patient characteristics.**

|  | Fluid inappropriate cohort | Fluid appropriate cohort | p |
|---|---|---|---|
| **Number of patients (%)** | 21 (68) | 10 (32) |  |
| **Mean arterial pressure, mmHg** | 66 [60, 75] | 63 [55, 65] | 0.23 |
| **Central venous pressure, mmHg** | 11 (4) | 12 (4) | 0.58 |
| **Mean pulmonary arterial pressure, mmHg** | 27 [25, 30] | 30 [23, 34] | 1.00 |
| **Pulmonary artery occlusion pressure, mmHg** | 17 [15, 22] | 15 [12, 16] | 0.14 |
| **Heart rate, beats per minute** | 86 (22) | 86 (25) | 0.99 |
| **Number of patients in sinus rhythm (%)** | 16 (76) | 8 (80.0) | 1.00 |
| **Urine output, ml/kg/h** | 0.9 [0.3, 1.4] | 0.6 [0.3, 0.9] | 0.65 |
| **Pulse Pressure Variation, %** | 5 [2, 9] | 4 [3, 13] | 0.57 |
| **Minimum IVC diameter, mm** | 16.6 (4.5) | 14.9 (4.2) | 0.34 |
| **Maximum IVC diameter, mm** | 21.2 (5.8) | 18.1 (3.6) | 0.15 |
| **Mixed venous oxygen saturation, %** | 74 [70, 76] | 66 [60, 75] | 0.19 |
| **Lactate, mmol/L** | 2.0 [1.1, 3.9] | 1.6 [1.1, 2.2] | 0.58 |
| **pH** | 7.32 (0.07) | 7.30 (0.15) | 0.73 |
| **paCO$_2$, mmHg** | 41 (7) | 42 (12) | 0.79 |
| **Cardiac Index, L/minute/m$^2$** | 3.1 (1.0) | 1.8 (0.3) | <0.001 |
| **GEDVI, ml/m$^2$** | 842 [678, 952] | 507 [460, 564] | 0.017 |
| **ELWI, ml/kg** | 14 [11, 16] | 11 [8, 15] | 0.50 |
| **PVPI** | 2.10 [1.97, 3.08] | 2.59 [1.89, 4.07] | 0.65 |
| **d.ETCO$_2$, %** | 1.5 [0.0, 2.0] | 1.0 [0.0, 2.0] | 0.98 |
| **Venous-to-arterial carbon dioxide difference, mmHg** | 5.0 [4.0, 7.4] | 6.0 [5.0, 6.8] | 0.61 |
| **Oxygen delivery index, ml/min/m2** | 390 [337, 523] | 248 [219, 276] | <0.001 |
| **Oxygen consumption index, ml/min/m2** | 107 (41) | 73 (17) | 0.017 |
| **IVC distensibility, %** | 24 (14) | 22 (16) | 0.75 |

Data are shown as count (%), median [1$^{st}$-3$^{rd}$ quartile] or mean (standard deviation). Abbreviations: GEDVI = global end diastolic volume index, ELWI = extravascular lung water index, PVPI = pulmonary vascular permeability index, d.E$_T$CO$_2$ = changes in E$_T$CO$_2$ during a passive leg raising test, IVC = inferior vena cava.

## Results

We enrolled 31 patients, for a total of 53 observations. Mean age was 68 (13) years and 77 (8) years in the fluid inappropriate and in the fluid appropriate group, respectively (p = 0.04), while male patients were 14 (67%) in the fluid inappropriate and 6 (60%) in the fluid appropriate group (p = 1). Fourteen patients were admitted to ICU for treatment of sepsis or septic shock, 10 patients for treatment of respiratory failure, 4 patients for post operative monitoring, two patients for post resuscitation care after cardiac arrest and one patient for treatment of traumatic hemorrhagic shock.

In Table 1, results of the first observation after enrollment for each patient are shown. In fourteen patients, CI was measured with PAC, while for seventeen patients PiCCO was used to calculate CI. For 10 patients, fluid administration was deemed appropriate, while for 21 patients it was deemed inappropriate. CVP was not significantly different between the two cohorts (p 0.58). Norepinephrine dose was, on median 0.31 [IQR 0.20, 0.66] micrograms/kg/min and 0.36 [0.10, 0.56] micrograms/kg/min in the **fluid inappropriate** (10 patients) and **fluid appropriate** cohort, respectively (p 0.56). In the **fluid inappropriate** cohort (8 patients), adrenaline dose was on median 0.25 [0.25, 0.25] micrograms/kg/min (1 patient) and dobutamine 7.50 [6.25, 8.75] micrograms/kg/min (2 patients).

**Table 2. Association between fluid appropriateness and study indices.**

| | Unadjusted | Adjusted |
|---|---|---|
| | Estimate (p value) | Estimate (p value) |
| **CVP** | 0.01 (0.94) | -0.2 (0.49) |
| **PPV** | -0.0006 (0.99) | -0.04 (0.48) |
| **d.E$_T$CO$_2$** | -0.09 (0.84) | -0.14 (0.71) |
| **IVC distensibility** | 0.01 (0.37) | 0.01 (0.62) |

Results from the linear mixed effects model.

Abbreviations: CVP = central venous pressure, PPV = pulse pressure variation, d.E$_T$CO$_2$ = changes in E$_T$CO$_2$ during a passive leg raising test, IVC = inferior vena cava.

Seven (30%) patients in the **fluid inappropriate** cohort had signs of fluid overload. No patients in the **fluid appropriate** cohort had signs of fluid overload, by definition. Using a PPV cutoff of 12% to define patients as fluid responders, we identified four fluid responder patients, i.e. two patients (20%) in the **fluid appropriate** cohort and two patients (10%) in the **fluid inappropriate** cohort (these two patients had no sign of fluid overload).

Tidal volume per kilogram of body weight was less than 8 ml/kg for all patients (on median, 5.7 [4.6–6.8] ml for the **fluid inappropriate** cohort and 5.3 [4.1–6.5] ml for the **fluid appropriate** cohort).

Results from the linear mixed effects model are shown in Table 2: there was no association between static and dynamic indices and fluid appropriateness.

## Discussion

This pilot study showed that there was no association between CVP, PPV, d.E$_T$CO$_2$, IVC distensibility and fluid appropriateness in our cohorts.

As has already been mentioned, the fact that a patient is a fluid responder does not mean that fluids are appropriate for the patient: from the hemodynamic point of view, fluid administration should be considered only when it is deemed useful in increasing a low cardiac output in presence of altered tissue perfusion and insufficient oxygen delivery, weighting benefits with risks associated to fluid overload [8, 10–12, 24]. Of note, the increase in cardiac output in fluid responder patients does not happen exclusively with fluid administration, but it can also be obtained with vasoconstrictors, which can act with a similar effect to fluids, i.e. by increasing preload [8, 25]. This is why we should ask ourselves, in front of a fluid responder patient, if fluid administration is justified (low cardiac output and no signs of fluid overload) or if a vasoconstrictor can be used (high cardiac output) [26]. Therefore, verifying if indices used to predict fluid responsiveness can identify **"fluid appropriate"** patients seems crucial, and to our knowledge this has not been evaluated to date.

It has been proved that CVP is not useful in determining if a patient is a fluid responder [7], and the results of this pilot study suggest that it is not useful in determining fluid appropriateness, either.

Dynamic indices, such as PPV, d.E$_T$CO$_2$, IVC distensibility perform better than static indices in evaluating fluid responsiveness, but they all suffer some limitations [27]. Moreover, they have mostly been studied in patients for whom fluids were probably not appropriate (i.e. patients with a high mean cardiac index or output [19, 28, 29]), therefore they should not be considered as indicators of fluid appropriateness, and this is confirmed by the results of our pilot study which show that, as for CVP, dynamic indices are not associated with fluid

appropriateness. Therefore we believe that haemodynamic monitoring's role in evaluating fluid appropriateness is, at the moment, irreplaceable.

The main limitation of this study, since our patients have a median PPV value of 5 [2, 9] % and of 4 [3, 13]% in the **fluid inappropriate** and in the **fluid appropriate** group, respectively, and a $d.E_TCO_2$ of less than 5% in both groups, our statement could be valid only for patients who are **not fluid responders**. In our cohorts, no patients were spontaneously breathing, and most patients were in sinus rhythm, but intra-abdominal pressure was not recorded and all patients were ventilated with a tidal volume of less than 8 ml/kg. Therefore, the fact that a small number of patients appear to be fluid responders when examining PPV and $d.E_TCO_2$ even if ⅓ of them are in the **fluid appropriate cohort** could be due to the intrinsic limitations of the indices we have described above. Even so, results should be confirmed in patients with a PPV ≥ 13%, and future studies in which intra-abdominal pressure is recorded and a "volume challenge" (done by increasing tidal volume to >8 ml/kg at the time of measurements) is performed [30] are warranted. Moreover, this is a single center pilot study, and the two cohorts are relatively small. Further research is needed to confirm these findings.

## Conclusions

Central venous pressure, pulse pressure variation, changes in $E_TCO_2$ during a passive leg raising test, inferior vena cava distensibility were not associated with fluid appropriateness in our cohorts. Hemodynamic monitoring is an invaluable tool to assess fluid appropriateness in critically ill patients.

## Author Contributions

**Conceptualization:** Giuseppe Natalini.

**Data curation:** Filippo Albani, Giuseppe Natalini.

**Formal analysis:** Filippo Albani, Giuseppe Natalini.

**Investigation:** Chiara Prezioso, Roberta Trotta, Erika Cavallo, Federica Fusina, Elena Malpetti, Filippo Albani, Rosalba Caserta, Antonio Rosano, Giuseppe Natalini.

**Methodology:** Giuseppe Natalini.

**Supervision:** Giuseppe Natalini.

**Validation:** Filippo Albani, Giuseppe Natalini.

**Writing – original draft:** Erika Cavallo, Federica Fusina.

**Writing – review & editing:** Federica Fusina, Antonio Rosano, Giuseppe Natalini.

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
