## [Decision Letter · Decision Letter 0]

27 Mar 2023

PONE-D-23-02768Central venous pressure and dynamic indices to assess fluid appropriateness in critically ill patientsPLOS ONE

Dear Dr. Fusina,

Thank you for submitting your manuscript to PLOS ONE. After careful consideration, we feel that it has merit but does not fully meet PLOS ONE’s publication criteria as it currently stands. Therefore, we invite you to submit a revised version of the manuscript that addresses the points raised during the review process.

We look forward to receiving your revised manuscript.

Kind regards,

Tatsushi Mutoh

Academic Editor

PLOS ONE

Journal Requirements:

Additional Editor Comments:

In view of the contents of this study, the article should be organized in a shorter form as a pilot study. 

Reviewers' comments:

Reviewer's Responses to Questions

**Comments to the Author**

1. Is the manuscript technically sound, and do the data support the conclusions?

Reviewer #1: Partly

Reviewer #2: Yes

2. Has the statistical analysis been performed appropriately and rigorously? 

Reviewer #1: Yes

Reviewer #2: I Don't Know

3. Have the authors made all data underlying the findings in their manuscript fully available?

Reviewer #1: Yes

Reviewer #2: Yes

4. Is the manuscript presented in an intelligible fashion and written in standard English?

Reviewer #1: No

Reviewer #2: Yes

5. Review Comments to the Author

Reviewer #1: This single center observational study aims to investigate the utility of CVP, PPV, IVC distensibility, and d.ETCO2 to identify fluid appropriateness in critically ill patients with various disease background. The authors concluded that those were not reliable indicator for fluid appropriateness.

Comments to the authors：

1. Methods (“Definition of fluid appropriateness”) and Discussion (“Therefore, the fact that　…described above.”) section:

The authors modified the diagnostic and therapeutic flow algorithm by Pinsky (Pinsky MR et al. 2005 Crit Care Med) and indicated the criteria of fluid appropriateness. In the discussion section, the authors stated that there were no patients appear to be fluid responders…are in the fluid appropriate cohort…are warranted. If so, although the authors considered as a limitation, research design can have profound negative influence on the outcomes and interpretation.

2. Kindly consider removing following sentences:

1) Methods section

We planned to determine…each index.

2) Results section

We did not undertake the…analyzed indices was significant.

Reviewer #2: This is a very interesting study which introduce a practical topic in which the authors evaluated several variables (CVP – PPV – and others) in the evaluation of fluid appropriateness. The aimed to provide a more deep vision than fluid responsiveness.

The main comment

The number of patients is very low to decide negative results. The sample size calculation is based on an assumption in which the standard deviation is much lower than that in the final results; Thus, the study seems underpowered and unable to provide significant results. The study should be revised to a pilot study which can be a hypothesis generating study for a larger well-powered study. I think the manuscript can be re-submitted in a shorter form (e.g., letter)

Abstract

You need to provide more details about the definition of “fluid appropriateness” and “fluid inappropriateness” in the methods. You can shorten the background to compensate for this. You can avoid mentioning all the names of the outcomes in the aim of the work and summarize them in one word.

There is no need to mention the names of statistical tests in the abstract.

You need more details in the results. Clarify the differences between the two groups (in brief) with regard to the main studied variables and then mention the predictive value (whether good or poor) for the main variables to detect fluid appropriateness.

Background

You need to decrease the number references especially in sentences which do not require to much references. (e.g., the first sentence in the article does not require 4 references, the last sentence in paragraph one does not require all these references)

Methods

There is no need to cite 4 references in the first two paragraphs in the protocol.

There is no need to cite another 3 references in the IVC examination.

Results

Please provide more details about the subdivisions of the patients. How many patients were deemed “fluid responders” but “fluid inappropriateness” and what were the causes of this discrepancy (e.g., high cardiac output, elevated filling pressures)

Discussion

The discussion should be more summarized and focused upon the current findings and there location in literature. There is no need to give full details about every variable and its pros and cons. You should go straight to the relation of your findings with any previous similar study.

6. PLOS authors have the option to publish the peer review history of their article (what does this mean?). If published, this will include your full peer review and any attached files.

Reviewer #1: No

Reviewer #2: **Yes: **Ahmed Hasanin

---

## [Author Response · Author response to Decision Letter 0]

3 May 2023

Dear Editor and Authors,

Thank you for reviewing our manuscript and for giving us the opportunity to revise it. We believe your comments have greatly improved the quality of our paper.

Here is a detailed, point by point reply to the comments provided.

Federica Fusina and the Authors

Journal Requirements:

ANSWER: We have modified our manuscript according to the Journal’s style requirements.

ANSWER: We have uploaded our anonymized data set at this address: https://github.com/filippo1985/afadysta.git

ANSWER: The ethics statement now appears in the Methods section only.

Additional Editor Comments:

In view of the contents of this study, the article should be organized in a shorter form as a pilot study. 

ANSWER: Thank you for your suggestion. We have modified the manuscript, which is now in the form of a pilot study. We have modified the title and manuscript to reflect this.

Reviewers' comments:

Reviewer's Responses to Questions

Comments to the Author

1. Is the manuscript technically sound, and do the data support the conclusions?

Reviewer #1: Partly

Reviewer #2: Yes

2. Has the statistical analysis been performed appropriately and rigorously? 

Reviewer #1: Yes

Reviewer #2: I Don't Know

3. Have the authors made all data underlying the findings in their manuscript fully available?

Reviewer #1: Yes

Reviewer #2: Yes

4. Is the manuscript presented in an intelligible fashion and written in standard English?

Reviewer #1: No

Reviewer #2: Yes

5. Review Comments to the Author

Reviewer #1: This single center observational study aims to investigate the utility of CVP, PPV, IVC distensibility, and d.ETCO2 to identify fluid appropriateness in critically ill patients with various disease background. The authors concluded that those were not reliable indicator for fluid appropriateness.

Comments to the authors：

1. Methods (“Definition of fluid appropriateness”) and Discussion (“Therefore, the fact that　…described above.”) section:

The authors modified the diagnostic and therapeutic flow algorithm by Pinsky (Pinsky MR et al. 2005 Crit Care Med) and indicated the criteria of fluid appropriateness. In the discussion section, the authors stated that there were no patients appear to be fluid responders…are in the fluid appropriate cohort…are warranted. If so, although the authors considered as a limitation, research design can have profound negative influence on the outcomes and interpretation.

ANSWER: Thank you for your comments. What we have reported is the mean PPV value. Using a PPV cutoff of 12% to define patients as fluid responders, we identified four fluid responder patients in our cohort (two in the fluid appropriate group and two in the fluid inappropriate group). We have added this information in the Results section of our manuscript, also incorporating a suggestion made by Reviewer 2: “Seven (30%) patients in the fluid inappropriate cohort had signs of fluid overload. No patients in the fluid appropriate cohort had signs of fluid overload, by definition. Using a PPV cutoff of 12% to define patients as fluid responders, we identified four fluid responder patients, i.e. two patients (20%) in the fluid appropriate cohort and two patients (10%) in the fluid inappropriate cohort (these two patients had no sign of fluid overload).”

2. Kindly consider removing following sentences:

1) Methods section

We planned to determine…each index.

2) Results section

We did not undertake the…analyzed indices was significant.

ANSWER: We have removed the sentences as requested.

Reviewer #2: This is a very interesting study which introduce a practical topic in which the authors evaluated several variables (CVP – PPV – and others) in the evaluation of fluid appropriateness. The aimed to provide a more deep vision than fluid responsiveness.

The main comment

The number of patients is very low to decide negative results. The sample size calculation is based on an assumption in which the standard deviation is much lower than that in the final results; Thus, the study seems underpowered and unable to provide significant results. The study should be revised to a pilot study which can be a hypothesis generating study for a larger well-powered study. I think the manuscript can be re-submitted in a shorter form (e.g., letter)

ANSWER: Thank you for your comments. You are right, we based our sample size calculation on data from a meta-analysis which had a lower standard deviation due to the higher number of patients analyzed, therefore our study is underpowered. The minimum difference in CVP between the fluid appropriate and fluid inappropriate group would need a very high sample size in order to obtain a high-powered study. In light of these comments, and as per Editor suggestion, we have modified our manuscript, which is now in the form of a pilot study. We have modified the title and manuscript accordingly.

Abstract

You need to provide more details about the definition of “fluid appropriateness” and “fluid inappropriateness” in the methods. You can shorten the background to compensate for this. You can avoid mentioning all the names of the outcomes in the aim of the work and summarize them in one word.

There is no need to mention the names of statistical tests in the abstract.

You need more details in the results. Clarify the differences between the two groups (in brief) with regard to the main studied variables and then mention the predictive value (whether good or poor) for the main variables to detect fluid appropriateness.

ANSWER: Thank you for your comments, we have modified the abstract accordingly. We did not add predictive values to the abstract since we did not undertake the calculation (in absence of significance) and, as requested by Reviewer 1, we completely removed this part from the methods and results sections. 

Background

You need to decrease the number references especially in sentences which do not require to much references. (e.g., the first sentence in the article does not require 4 references, the last sentence in paragraph one does not require all these references). 

ANSWER: We have reduced the number of references from 51 references to 30, leaving only the most significant ones. 

Methods

There is no need to cite 4 references in the first two paragraphs in the protocol.

There is no need to cite another 3 references in the IVC examination.

ANSWER: We have reduced the number of references from 51 references to 30, leaving only the most significant ones. 

Results

Please provide more details about the subdivisions of the patients. How many patients were deemed “fluid responders” but “fluid inappropriateness” and what were the causes of this discrepancy (e.g., high cardiac output, elevated filling pressures) 

ANSWER: Thank you for your comments. Patients with fluid overload (defined as PAOP > 18 mmHg or GEDVI > 800 ml/m² and ELWI > 10 ml/Kg were 7 (30%) in the fluid inappropriate group. No patients in the fluid appropriate group had signs of fluid overload, by definition. Using a PPV cutoff of 12% to define patients as fluid responders, we identified four fluid responder patients in our cohort, two in the fluid appropriate group and two in the fluid inappropriate group (these two patients had no sign of fluid overload). We have added this information in the Results section of our manuscript, also incorporating a suggestion made by Reviewer 1: “Seven (30%) patients in the fluid inappropriate cohort had signs of fluid overload. No patients in the fluid appropriate cohort had signs of fluid overload, by definition. Using a PPV cutoff of 12% to define patients as fluid responders, we identified four fluid responder patients, i.e. two patients (20%) in the fluid appropriate cohort and two patients (10%) in the fluid inappropriate cohort (these two patients had no sign of fluid overload).”

Discussion

The discussion should be more summarized and focused upon the current findings and there location in literature. There is no need to give full details about every variable and its pros and cons. You should go straight to the relation of your findings with any previous similar study.

ANSWER: Thank you for your suggestion. We have modified the discussion in order to reflect your comment. 

6. PLOS authors have the option to publish the peer review history of their article (what does this mean?). If published, this will include your full peer review and any attached files.

Do you want your identity to be public for this peer review? For information about this choice, including consent withdrawal, please see our Privacy Policy.

Reviewer #1: No

Reviewer #2: Yes: Ahmed Hasanin

---

## [Editor Report · Decision Letter 1]

5 May 2023

Central venous pressure and dynamic indices to assess fluid appropriateness in critically ill patients: a pilot study

PONE-D-23-02768R1

Dear Dr. Fusina,

We’re pleased to inform you that your manuscript has been judged scientifically suitable for publication and will be formally accepted for publication once it meets all outstanding technical requirements.

Kind regards,

Tatsushi Mutoh

Academic Editor

PLOS ONE
---

## [Editor Report · Acceptance letter]

9 May 2023

PONE-D-23-02768R1 

Central venous pressure and dynamic indices to assess fluid appropriateness in critically ill patients: a pilot study 

Dear Dr. Fusina:

I'm pleased to inform you that your manuscript has been deemed suitable for publication in PLOS ONE. Congratulations! Your manuscript is now with our production department. 

Kind regards, 

on behalf of

Dr. Tatsushi Mutoh 

Academic Editor

PLOS ONE